# OpenReview forum: "The Jailbreak Tax: How Useful are Your Jailbreak Outputs?"
_ICLR.cc/2025/Workshop/BuildingTrust — BuildingTrust_

### Official Review · Reviewer_CNDd · 2025-02-16
**This is a great paper!**

**Rating:** 9
**Confidence:** 4

**Review:**

# Summary
The paper introduces the concept of a "jailbreak tax," representing the degradation in model performance when a jailbreak attack successfully circumvents alignment guardrails. This work addresses a critical gap in jailbreak evaluations: existing research often focuses on whether the jailbreak bypasses safety mechanisms but overlooks the utility or correctness of the resulting outputs. The authors construct a novel evaluation methodology by aligning models to refuse benign topics like math and biology and then assessing the correctness of jailbreak outputs against these pseudo-harmful tasks. The experiments span eight jailbreak methods, multiple alignment approaches (system prompt, fine-tuning, harmful task mixing), and three models, revealing substantial variance in jailbreak tax across methods.

## Strengths

- Evaluating the utility of jailbreak outputs is a critical but underexplored area in AI safety research. This paper takes a meaningful first step toward addressing this gap.

- The use of pseudo-harmful tasks (e.g., EvilMath) allows for objective evaluation of jailbreak correctness. The authors employ multiple alignment techniques and jailbreak methods, providing a robust empirical analysis.

- The authors systematically analyze the relationship between jailbreak success and utility degradation, identifying methods (e.g., Many-shot) that preserve performance better than others (e.g., PAIR).

## Limitations

- While the pseudo-harmful task approach is a clever methodological choice, it is not clear how well the results translate to real-world jailbreak scenarios involving genuinely harmful content (e.g., bomb-making instructions or phishing attacks). This is an inherently difficult problem to measure, but the paper could benefit from a more explicit discussion of this limitation.

- The evaluation tasks (math, biology multiple-choice) are relatively well-structured with clear ground truth answers. However, real-world jailbreak queries often involve complex, multi-step reasoning or subjective judgments (e.g., generating persuasive phishing emails or constructing realistic social engineering scenarios). Evaluating utility in such open-ended tasks may introduce additional challenges that are not fully captured by the current methodology. While this is a difficult problem, this paper represents a valuable foundational step towards it.

Overall, a great contribution!

---

### Official Review · Reviewer_2t2A · 2025-02-28
**An interesting investigation about jailbreaking tax**

**Rating:** 7
**Confidence:** 4

**Review:**

This work provides a quantified framework for investigating the 'real' success of a jailbreaking test instead of the rate of non-refusal responses. The motivation is practical and results show that a number of jailbreak methods achieves low jailbreak tax (i.e. the real success) even though they achieve high successful jailbreak rate.

pros:
1. The authors employ a well-structured framework and evaluating jailbreak techniques across multiple benchmarks, which allows for a systematic and measurable evaluation of the jailbreak tax.
2. The use of datasets including GSM8K, MATH, WMDP ensures that the results is well-quantified.

cons:
1. the score of 'utility' definition is limited. The paper define utility with correctness on benchmark tasks, which may not fully capture real-world risks. For example, an incorrect response to a biological weapon-related query could still provide dangerous information while no ground truth is provided.

follow-up question is that could the jailbreak tax generalized to open-ended malicious questions where no ground-truth exists? Open-ended question answering is a more generalizable scene for AI safety from my perspective.

---

### Official Review · Reviewer_t62Q · 2025-02-28
**Novel and well-executed work on assessing risks of jailbroken language models**

**Rating:** 7
**Confidence:** 3

**Review:**

## Summary
This paper introduces the concept of a **jailbreak tax** — a measure of how much model utility degrades when jailbreak techniques bypass safety guardrails. The authors propose a novel evaluation framework that avoids the challenges of evaluating real-world dangerous outputs by using benign, easily-verified tasks as proxies. They align models to refuse these benign tasks using several alignment techniques, apply various jailbreak techniques, and then measure the degradation in utility compared to unaligned models.

---

## Strengths

**Clear and Well-Motivated Core Idea**
The authors clearly put forward the case that while previous work has focused primarily on jailbreak success rate (i.e., whether models can be made to respond to harmful queries), the question of how these jailbreaks affect the model’s capabilities remains understudied. Their introduction of a jailbreak tax as a metric for quantifying capability degradation during jailbreaking provides a useful framework for evaluating the true effectiveness of different attack methods.

**Evaluation Framework**
They propose a clever evaluation framework for measuring the utility of jailbroken models by getting models to refuse questions with known ground-truth answers, thereby avoiding challenges associated with evaluating genuinely harmful content.

**Comprehensive Experimental Design**
The paper covers a good range of jailbreak methods across three model sizes and multiple alignment techniques. The inclusion of the UnicornMath dataset as a control for out-of-distribution effects demonstrates careful attention to experimental design.

**Interesting Results**
The paper provides several interesting findings that have the potential to impact the jailbreak mitigation work of language model developers:

- **The lack of correlation between jailbreak success rate and jailbreak tax**
  The authors note that even when a jailbreak technique frequently succeeds in eliciting a response, it does not necessarily impose a high utility cost on the model.

- **Larger models do not reduce the jailbreak tax**
  The results suggest that scaling model size does not inherently mitigate the utility losses imposed by jailbreaks.

- **Harder tasks incur higher jailbreak tax**
  They demonstrate that more challenging queries lead to greater performance degradation when the model is jailbroken.

**Clear Presentation**
The paper is clear and well-written.

---

## Weaknesses

**Lacking Comparison to Prior Work**
While the paper references other jailbreak benchmarks such as StrongREJECT, it would be helpful if the authors provided a more explicit comparison of their findings with prior work. For instance, Figure 4 of StrongREJECT suggests a correlation between the rate of non-refusal and model capability, whereas this paper asserts that there is “no apparent correlation.”

**Potential Contradiction in the LLaMA 405B Results**
The authors conclude that “there is no apparent correlation between a jailbreak’s success rate and its impact on model utility,” yet for the LLaMA 405B experiments, the data seems to show a relatively strong correlation. Is this a spurious correlation or does a stronger correlation emerge as models get larger/more capable?

**Highlighting the Low-Tax Jailbreak Cases**
A notable finding of this work is that for all of the alignment techniques considered, there are existing jailbreaks with low (~0-10%) jailbreak tax. It would be good if this finding were highlighted in the text.

---

### Decision · Program_Chairs · 2025-03-04

Accept